# Flexible Ag$_2$Se-based thin-film thermoelectrics for sustainable energy harvesting and cooling

Wenyi Chen[1,2], Meng Li[1], Xiaodong Wang[3], Joseph Otte[4], Min Zhang[1], Chengyang Zhang[1], Tianyi Cao[1], Boxuan Hu[1], Nanhai Li[1], Wei-Di Liu[1], Matthew Dargusch[2], Jin Zou[2,4], Qiang Sun[5] ✉, Zhi-Gang Chen[1] ✉ & Xiao-Lei Shi[1] ✉

The high cost and complexity of fabrication limit the large-scale application of flexible inorganic thermoelectric materials. Currently, Bi$_2$Te$_3$-based materials are the only commercially viable option, but the inclusion of Te significantly increases production costs. This study presents a simple and cost-effective method for fabricating flexible Ag$_2$Se films, employing a combination of solvothermal synthesis, screen printing, and spark plasma sintering. The incorporation of a small amount of Te improves film density and facilitates Te diffusion doping, leading to Ag$_2$Se films with a high power factor of 25.7 μW cm$^{-1}$ K$^{-2}$ and a figure of merit (*ZT*) of 1.06 at 303 K. These films exhibit excellent flexibility, retaining 96% of their performance after 1000 bending cycles at a 5 mm bending radius. Additionally, we design a flexible thermoelectric device featuring a triangular p-n junction structure based on these films. This device achieves a normalized power density of 4.8 μW cm$^{-2}$ K$^{-2}$ at a temperature difference of 20 K and a maximum cooling of 29.8 K with an input current of 92.4 mA. These findings highlight the potential of this fabrication method for developing thermoelectric materials and devices for energy harvesting and cooling applications.

Thermoelectric technology, especially flexible thermoelectric devices (F-TEDs), is important for converting heat into electricity for power generation and cooling[1,2]. These devices are well-suited for powering and controlling the temperature of small electronics like smartphones and smartwatches[3,4]. To achieve sufficient power and cooling performance, F-TEDs require thermoelectric materials with a high power factor (*S$^2$σ*) and figure of merit (*ZT*) at near-room temperatures[4,5]. The *ZT* value is given by *ZT* = *S$^2$σT/κ*, where *S* is the Seebeck coefficient, *σ* is

electrical conductivity, *T* is absolute temperature, and *κ* is thermal conductivity[2]. The *κ* includes both lattice thermal conductivity (*κ$_l$*) and electronic thermal conductivity (*κ$_e$*)[2]. However, most inorganic materials with good thermoelectric properties lack flexibility, while organic materials, though flexible, usually have poor thermoelectric performance. Therefore, researchers are working to improve the flexibility of inorganic thermoelectric materials, as these materials are more advanced in development[6,7].

[1]School of Chemistry and Physics, ARC Research Hub in Zero-emission Power Generation for Carbon Neutrality, and Centre for Materials Science, Queensland University of Technology, Brisbane, QLD, Australia. [2]School of Mechanical and Mining Engineering, The University of Queensland, Brisbane, QLD, Australia. [3]Central Analytical Research Facility, Institute for Future Environments, Queensland University of Technology, Brisbane, QLD, Australia. [4]Centre for Microscopy and Microanalysis, The University of Queensland, Brisbane, QLD, Australia. [5]State Key Laboratory of Oral Diseases & National Center for Stomatology & National Clinical Research Center for Oral Diseases, West China Hospital of Stomatology, Sichuan University, Chengdu, Sichuan, PR China. ✉e-mail: qiangsun@scu.du.cn; zhigang.chen@qut.edu.au; xiaolei.shi@qut.edu.au

Among thermoelectric materials, $Bi_2Te_3$ has the best performance near room temperature and is the only material used in commercial applications[8]. However, its reliance on tellurium (Te) makes it expensive and limits large-scale use[9]. $Ag_2Se$ is a narrow band-gap n-type semiconductor that exists as an orthorhombic phase ($\beta$-$Ag_2Se$) at room temperature and transforms into a cubic phase ($\alpha$-$Ag_2Se$) at high temperatures ($\approx$130 °C)[10]. $\beta$-$Ag_2Se$ has thermoelectric properties comparable to $Bi_2Te_3$ due to its high carrier mobility ($\mu$) and low $\kappa_l$, making it a promising alternative[2,3]. Recent studies on bulk $Ag_2Se$ have demonstrated that preferentially fabricating $Ag_2Se$ crystals through one-step sintering can enhance the $ZT$ to 0.95 at 300 K[2]. Additionally, using a solution-based method and optimizing the sintering temperature can increase $ZT$ to 1.1 at 373 K[11]. Furthermore, doping $Ag_2Se$ with 0.2 at.% Zn has been shown to further improve $ZT$ to 1.3 at 393 K[12]. More recently, 3D-printed $Ag_2Se$ pellets achieved a $ZT$ of 1.3 at room temperature, further highlighting its potential[13]. However, challenges such as high porosity ($\approx$50%), excessive thickness ($\approx$0.8 mm), and limited flexibility hinder its use in wearable and microelectronic devices. Moreover, bulk $Ag_2Se$ does not fully utilize its crystal structure, which provides greater ductility than $Bi_2Te_3$, making it a strong candidate for flexible thermoelectric materials[2].

Several methods have been developed to fabricate $Ag_2Se$ films, including magnetron sputtering[14], vacuum filtration[3], co-evaporation[5], physical vapor deposition (PVD)[15], and screen printing[16]. However, achieving a $ZT$ exceeding 1 remains challenging across these techniques. Among these, screen printing is the most straightforward, cost-effective, and efficient. Despite its advantages, most screen-printed $Ag_2Se$ films show poor thermoelectric performance[17,18]. This is mainly due to their high porosity, which reduces $\sigma$ and lowers the $S^2\sigma$[17-19]. Addressing this issue while maintaining flexibility remains a key challenge in optimizing screen-printed $Ag_2Se$ films. Several strategies have been explored to improve the thermoelectric performance of $Ag_2Se$ films, including film orientation control[20], doping[5,21], sandwich engineering[22], and the addition of secondary phases[5]. Some studies suggest that enhancing the (013) orientation can increase $\mu$ and $\sigma$, leading to a higher $S^2\sigma$[5]. However, a significant increase in $\sigma$ can also raise $\kappa_e$, which may reduce the $ZT$[5]. Additionally, since $Ag_2Se$ lacks a layered crystal structure, controlling its orientation is challenging[14]. Traditional doping methods also have limitations. $Ag_2Se$ has a high intrinsic carrier concentration ($n$) due to the low formation energy of selenium (Se) vacancies[5]. Reducing $n$ could improve the $S$, but cation doping usually increases $n$, which lowers $S$, as Ag exists in the +1 valence state[5]. Therefore, anion doping and the introduction of secondary phases are considered more effective strategies[3,5]. However, anion doping, such as sulphur doping, can alter the primary phase and reduce $ZT$ in some cases[23]. The addition of secondary phases is also challenging, as selecting a suitable phase to enhance $Ag_2Se$'s properties is not straightforward[14]. Research on these approaches, especially for flexible films, remains limited, highlighting opportunities for further study.

## Result and discussion

Most inorganic films produced by screen printing have low density due to high porosity[17,18]. To address this, reducing particle size in the ink or adding fillers can improve film densification[17,19,24]. Our previous study[18] showed that Te acts as a nanobinder, enhancing the thermoelectric properties of screen-printed $Bi_2Te_3$ films. We also fabricated $Ag_2Se$ films in that study, confirming the effectiveness of Te in screen-printed films. However, at the time, research on $Ag_2Se$ was still in its early stages, and the role of Te in $Ag_2Se$ films remained unclear. Further investigation is needed to determine the optimal Te content and its impact on $Ag_2Se$ films. In this work, we synthesized $Ag_2Se$ microparticles and Te nanorods (Supplementary Figs. 1 and 2) using solvothermal methods, followed by screen printing and spark plasma sintering (SPS) to produce a dense, flexible $Ag_2Se$/Te composite film.

Unlike our previous findings, we observed that Te in $Ag_2Se$ films undergoes diffusion doping during annealing, optimizing the bandgap. The unit-cell structures of pure and Te-doped $Ag_2Se$ are shown in Supplementary Figs. 3, and Fig. 1a illustrates the mechanism of Te incorporation. Te acts as a filler, increasing film density and enhancing $\mu$ and $\sigma$. Additionally, the Te phase induces an energy filtering effect, optimizing both $n$ and the $S$. This diffusion doping increases the $Ag_2Se$ bandgap from 0.07 to 0.127 eV, further improving $n$ and $S$. As a result, the $Ag_2Se$/Te film achieves an exceptional $S^2\sigma$ of 25.7 $\mu$W cm$^{-1}$ K$^{-2}$ at 303 K, the highest reported for screen-printed films (Fig. 1b)[16,25-41]. The soldering effect of Te nanorods also enhances film flexibility, as shown in the inset. To evaluate practical applications, we designed a flexible thermoelectric device (F-TED) with a triangular p-n junction, consisting of 10 pairs of p-type $Bi_{0.4}Sb_{1.6}Te_3$ legs and n-type $Ag_2Se$/Te legs, as shown in Fig. 1c. The device achieves a power density ($\omega$) of 1.9 mW cm$^{-2}$ at a temperature difference ($\Delta T$) of 20 K, corresponding to a normalized power density ($\omega_n$) of 4.8 $\mu$W cm$^{-2}$ K$^{-2}$, comparable to devices made using other fabrication methods (Supplementary Table 1 and Fig. 1d)[5,9,16,25-28,35,41-85]. The Peltier effect was also evaluated, showing a maximum cooling performance ($\Delta T_{max}$) of 29.8 K under a 92.4 mA current without external heat sinks. These results highlight the potential of the $Ag_2Se$/Te film for power generation and thermal management applications.

To determine the optimal amount of Te for enhancing the thermoelectric performance of $Ag_2Se$ films, we synthesized a series of $Ag_2Se + x$Te compositions, where $x = 0$, 2.5, 5, 7.5, and 10 wt.%. X-ray diffraction (XRD) was used to examine the phase structure of the resulting films. Figure 2a shows the XRD patterns in the $2\theta$ range from 20° to 55°. All major diffraction peaks align with the orthorhombic $Ag_2Se$ phase (PDF #04-002-0445) and hexagonal Te phase (PDF #04-027-7719). As the Te content increases, the intensities of Te-related diffraction peaks strengthen, particularly displaying a (100) preferred orientation, likely due to the pressure applied during SPS. For a more detailed analysis, Rietveld refinement of the XRD data was performed for films with 0, 5, and 10 wt.% Te, as shown in Fig. 2b. Additional refinements for films with 2.5 and 7.5 wt.% Te are presented in Supplementary Fig. 4, with detailed results provided in Supplementary Table 2. The refinement results reveal that as the Te content increases, the lattice parameters ($a$, $b$, $c$) of $Ag_2Se$ expand, suggesting that $Te^{2-}$ ions (97 pm) replace $Se^{2-}$ ions (50 pm), as depicted in Fig. 2c. Figure 2d shows the full X-ray photoelectron spectroscopy (XPS) spectrum of the $Ag_2Se$ film with 5 wt.% Te, confirming the presence of Ag, Se, and Te. Figure 2e provides high-resolution scans of Te $3d_{5/2}$, revealing three distinct states of Te in the $Ag_2Se$ film: $Te^0$ (573 eV), $Te^{2-}$ (576.1 eV), and $Te^{4+}$ (572.6 eV). These findings support both the XRD and Rietveld refinement results, further confirming the diffusion of Te into the $Ag_2Se$ structure.

The microstructure and composition of the as-fabricated $Ag_2Se$/Te films were analyzed using scanning electron microscopy (SEM) combined with energy-dispersive X-ray spectroscopy (EDS). Figure 2f, g presents both top and cross-sectional SEM images of $Ag_2Se$ films with 0 and 5 wt.% Te. The top-view images reveal that adding Te significantly improves the densification of the $Ag_2Se$ film. As the Te content increases, the film thickness remains approximately constant at around 15 $\mu$m, as shown in the cross-sectional views. Additional SEM images of $Ag_2Se$ films with varying compositions are provided in Supplementary Figs. 5 and 6. The film thickness and uniformity show only slight variation (12–16 $\mu$m) with different Te content since all films were fabricated using 160 mesh screens. This consistency is attributed to two factors: First, the Te content in the precursor is relatively low compared to $Ag_2Se$. Second, all films were annealed under the same SPS conditions, which involved high temperature and pressure, ensuring uniform densification and minimal influence of additional Te on film morphology. Further analysis of the $Ag_2Se$ film with 5 wt.% Te, including secondary electron (SE) and backscattered electron (BSE)

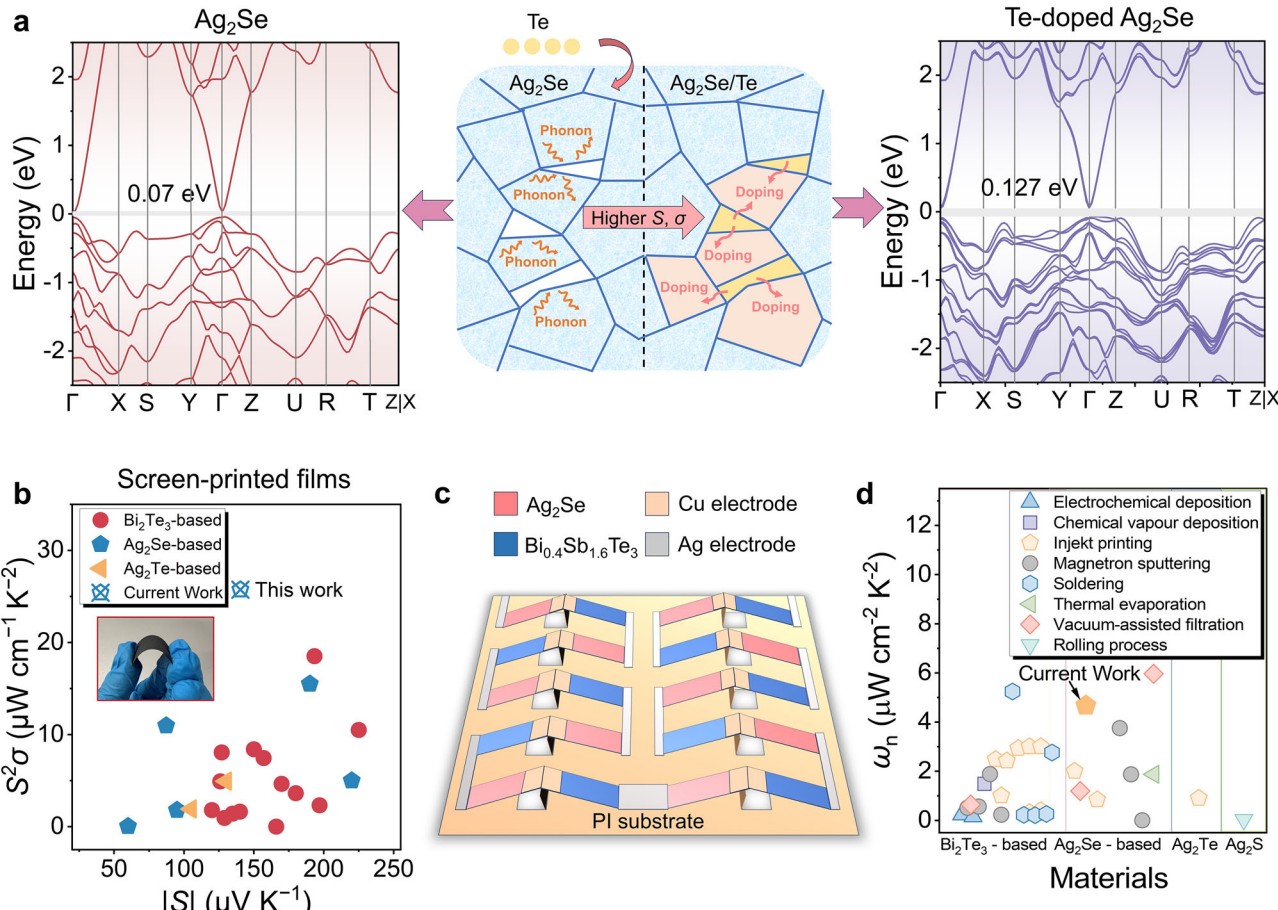

**Fig. 1 | Overview of screen-printed flexible Ag$_2$Se/Te composite films and their thermoelectric devices. a** Schematic illustration of the mechanism and structure of screen-printed Ag$_2$Se/Te films. The right side shows the calculated bandgap of pure Ag$_2$Se, while the left side shows the bandgap of Te-doped Ag$_2$Se.
**b** Comparison of power factor ($S^2\sigma$) as a function of the Seebeck coefficient ($S$) between the Ag$_2$Se film developed in this work and previously reported near-room-temperature inorganic thin films fabricated by screen printing[16,25-41]. The inset shows the bending flexibility of the Ag$_2$Se film. **c** Schematic of the thermoelectric device structure designed in this study. **d** Comparison of normalized power density ($\omega_n$) among thermoelectric devices produced using different fabrication methods[5,9,16,25-28,35,41-85]. Inkjet printing includes spin coating, machine printing, blade printing, and screen printing.

images, along with the corresponding EDS maps, is shown in Fig. 2h, i. These results confirm the presence of Ag, Se, and Te elements. Te nanorods are observed filling the gaps between the Ag$_2$Se grains, effectively reducing the porosity of the film, which aligns with our previous findings[18]. Additional SEM and EDS results for films with different Te contents are available in Supplementary Figs. 7–9 for reference.

To examine the nanostructure of the as-fabricated Ag$_2$Se/Te film, particularly the Te-doped regions, we used the focused ion beam (FIB) technique to prepare a lamella specimen of Ag$_2$Se with 5 wt.% Te. Transmission electron microscopy (TEM) was then performed to characterize the sample. Figure 3a shows a low-magnification scanning transmission electron microscopy (STEM) image of the lamella sample. Figure 3b presents an enlarged high-angle annular dark-field (HAADF) STEM image along with EDS maps for the individual Ag, Se, and Te elements, derived from a selected area in Fig. 3a. These images reveal a uniform distribution of Ag, Se, and Te, indicating homogeneous Te doping (diffusion) throughout the sample. Figure 3c displays a high-magnification, spherical aberration-corrected scanning TEM (Cs-STEM) HAADF image, with insets showing a magnified Cs-STEM HAADF image and a fast Fourier transform (FFT) pattern. This confirms that the image was captured along the [0 1 0] zone axis of the Ag$_2$Se matrix, demonstrating a well-ordered lattice structure and confirming the high crystallinity of the prepared film. Figure 3d presents a

magnified high-resolution TEM (HRTEM) image from the same region, showing lattice contrast that indicates lattice distortion. This distortion is attributed to the substitution of Se$^{2-}$ ions by Te$^{2-}$ ions within the Ag$_2$Se lattice, which aligns with the results from the XRD Rietveld refinement. Figure 3e, f shows low- and high-magnification HRTEM images of the Ag$_2$Se matrix, taken from an area without noticeable lattice contrast, as seen in Fig. 3d. The inset shows the corresponding selected area electron diffraction (SAED) pattern with indexed information, confirming the imaging direction is along the [0 1 0] zone axis of Ag$_2$Se. Figure 3g displays a high-magnification HRTEM image of a selected area with lattice contrast, and Fig. 3h shows corresponding strain maps along various directions. These maps indicate that the strain is primarily along the x-direction, caused by the lattice distortion resulting from the Te$^{2-}$ substitution in the Ag$_2$Se lattice. Lastly, Fig. 3i presents inverse Fourier transform images of Fig. 3g, revealing the presence of potential edge-like dislocations. The inset provides an enlarged view of these dislocations, which are likely caused by the substitution of Se$^{2-}$ by Te$^{2-}$ within the Ag$_2$Se lattice.

To examine the impact of Te content $x$ ($x = 0$, 2.5, 5, 7.5, and 10 wt.%) on the thermoelectric performance of Ag$_2$Se films, we measured their properties over a temperature range of 303 K to 383 K. Figure 4a–c presents the temperature-dependent $S$, $\sigma$, and $S^2\sigma$ of Ag$_2$Se/Te films. As $x$ increases from 0 to 5 wt.%, both $S$ and $\sigma$ improve, resulting in an optimized $S^2\sigma$ of 25.7 μW cm$^{-1}$ K$^{-2}$ at 303 K.

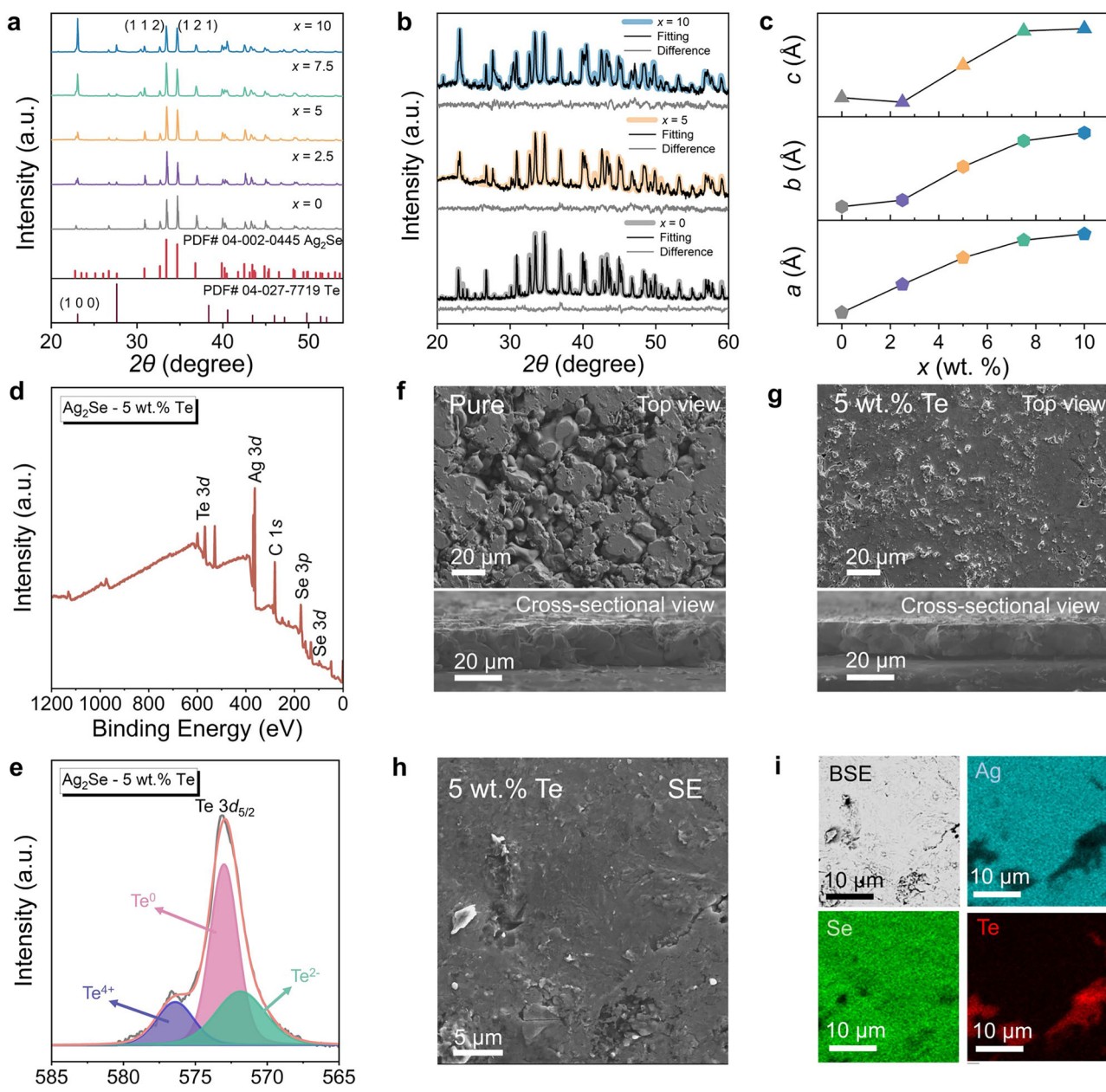

**Fig. 2 | Phase and microstructure of screen-printed Ag₂Se/Te films. a** X-ray diffraction (XRD) patterns of Ag₂Se thin films with different Te contents ($x = 0$, 2.5, 5, 7.5, and 10 wt.%). **b** Rietveld quantitative phase analysis of Ag₂Se thin films with $x = 0$, 5, and 10 wt.% Te. **c** Corresponding lattice parameters of Ag₂Se with $x = 0$, 2.5, 5, 7.5, and 10 wt.% Te. **d** Full X-ray photoelectron spectroscopy (XPS) survey of Ag₂Se film with $x = 5$ wt.%. **e** Detailed XPS survey for Te 3$d$ of Ag₂Se film with $x = 5$ wt.% Te. Scanning electron microscopy (SEM) images of Ag₂Se films with **f** $x = 0$ and **g** $x = 5$ wt.% Te from both top and cross-sectional views. **h** Secondary electron (SE) SEM images of Ag₂Se film with $x = 5$ wt.% Te from 565 to 585 eV. **i** Corresponding backscattered electron (BSE) SEM images and energy dispersive X-ray spectroscopy (EDS) maps for Ag, Se, and Te.

However, when $x$ increases from 7.5 to 10 wt.%, excessive formation of the secondary Te phase within the Ag₂Se film leads to a decline in $\sigma$, counteracting the increase in $S$ and reducing $S^2\sigma$. To further understand the variations in $S$ and $\sigma$ at room temperature, we conducted Hall measurements to evaluate $n$ and $\mu$ as functions of $x$ (Fig. 4d). As $x$ increases from 0 to 5 wt.%, $\mu$ gradually improves because Te fills the pores in the Ag₂Se film. However, when $x$ increases from 7.5 to 10 wt.%, the excessive Te phase enhances phonon scattering, leading to a reduction in $\mu$. Regarding $n$, it decreases as $x$ increases from 0 to 10 wt.%. This trend, along with the increase in $S$, can be explained by the calculated bandgap changes

based on first-principles DFT calculations (Supplementary Fig. 10): Ag₂Se has a bandgap of 0.07 eV, while Te-doped Ag₂Se exhibits a slightly larger bandgap of 0.127 eV. The widening bandgap after Te doping makes it more difficult for electrons to transition to the conduction band, resulting in lower $n$ and higher $S$.

To better understand the variations in $\mu$ and $n$ with different Te contents, Fig. 4e compares the effective mass ($m^*$) and deformation potential ($E_{def}$) as functions of $x$, based on the single parabolic band (SPB) model. As $x$ increases from 0 to 7.5 wt.%, $m^*$ gradually rises and then stabilizes. This trend may be attributed to the formation of a secondary Te phase within Ag₂Se. The phase boundaries between Te and Ag₂Se can

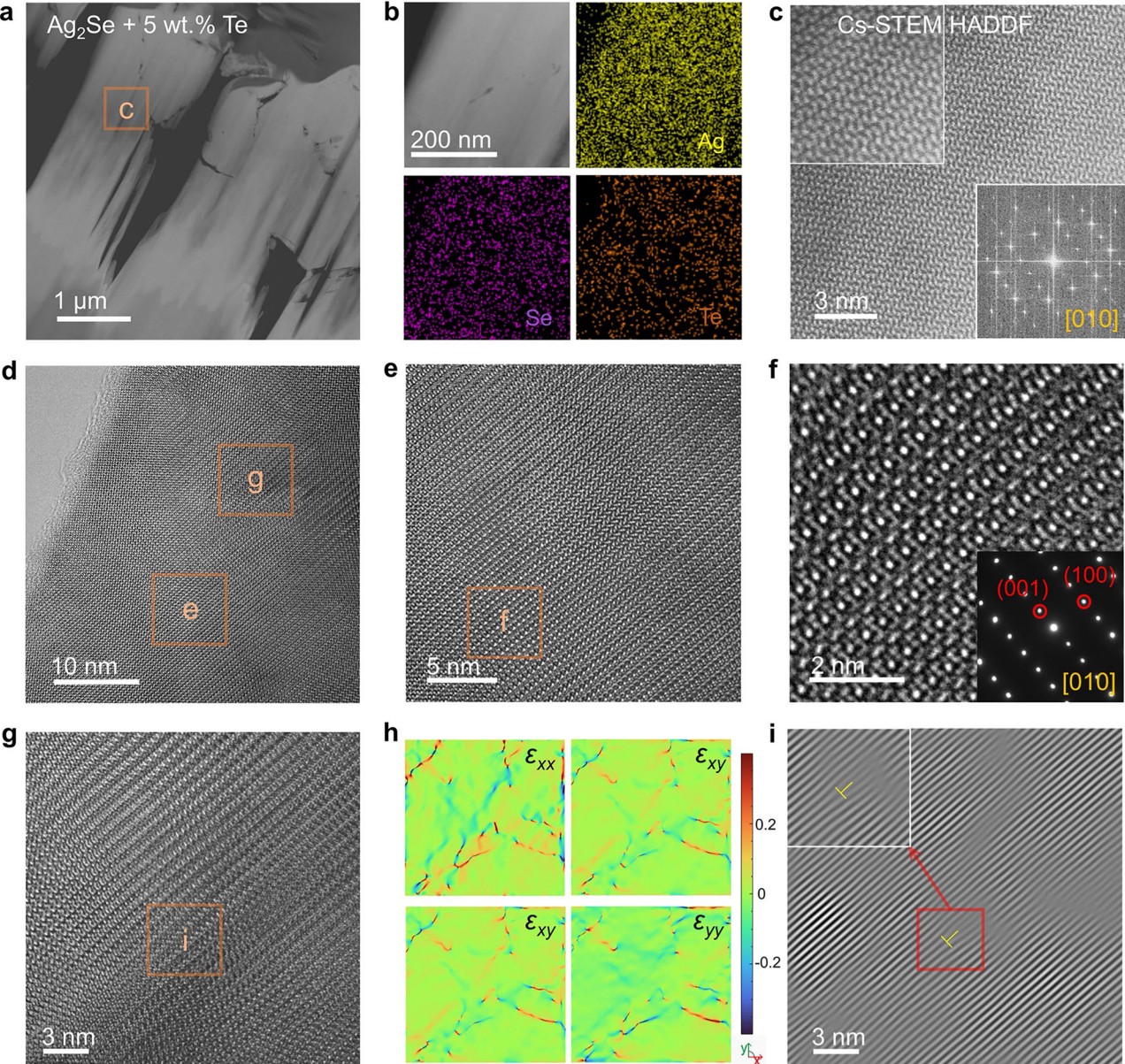

**Fig. 3 | Characterizations of micro/nanostructures for screen-printed Ag₂Se films with 5 wt.% Te. a** Low-magnification high-angle annular dark-field (HAADF) image of the specimen fabricated by focused ion beam (FIB) technique. **b** Enlarged HAADF image and its corresponding EDS maps for Ag, Se, and Te elements. **c** Magnified spherical aberration-corrected scanning TEM (Cs-STEM) HADDF image taken from the selected area of (**a**). The inset shows the corresponding fast Fourier transform (FFT) pattern with the view zone axis of [0 1 0]. **d** Low-magnification TEM image derived from (**a**). **e** High-resolution TEM (HRTEM) image taken from (**d**). **f** Magnified HRTEM image of the Ag₂Se matrix. The inset shows the corresponding selected area electron diffraction (SAED) pattern viewing along the (0 1 0) direction. **g** HRTEM image with potential lattice imperfections taken from (**d**). **h** Corresponding strain maps along different directions. **i** Inverse Fourier transform image taken from (**g**) presents the potential presence of edge-like dislocations. The inset shows a magnified area with a potential edge-like dislocation.

induce an energy filtering effect, selectively blocking low-energy carriers, and enhancing the $S$. To confirm the presence of this mechanism, we performed first-principles DFT calculations (Supplementary Fig. 10). The results show that the bandgap of Te is nearly 0 eV, while Ag₂Se has a bandgap of 0.07 eV, and Te-doped Ag₂Se exhibits an increased bandgap of 0.127 eV. The significant differences in band structures among Te, Ag₂Se, and Te-doped Ag₂Se create an effective energy barrier, facilitating the energy filtering effect. When $x$ increases from 7.5 to 10 wt.%, $m^*$ decreases due to the suppression of $n$, as the secondary Te phase within Ag₂Se becomes saturated. Regarding $E_{def}$, it initially decreases with $x$ increase from 0 to 5 wt.% since Te atoms fill gaps within the Ag₂Se matrix, altering the material's overall deformation behavior, consistent with previous research[18]. However, as $x$ increases from 5 to 7.5 wt.%, $E_{def}$

gradually rises, indicating reduced lattice deformability. This suggests that Te doping becomes more dominant once the Te content within Ag₂Se reaches saturation. Figure 4f compares the $S^2\sigma$ predicted by the SPB model with measured $S^2\sigma$ as a function of $n$, demonstrating that Te doping effectively optimizes $n$ to its optimal range.

To evaluate the thermal transport properties of Ag₂Se films with varying Te content ($x = 0, 2.5, 5, 7.5,$ and 10 wt.%), we measured the in-plane thermal diffusivity ($D$) using the photothermal intensity technique (PIT) with an alternating current (AC) method. The $\kappa$ was then calculated using the equation $\kappa = D \times C_p \times \rho$, where $\rho$ is the mass density (determined via the specific gravity method) and $C_p$ is the specific heat capacity. The measured $D$ values are shown in Supplementary Fig. 11, while Fig. 4g presents $\kappa$ as a function of $x$ at room temperature. As $x$

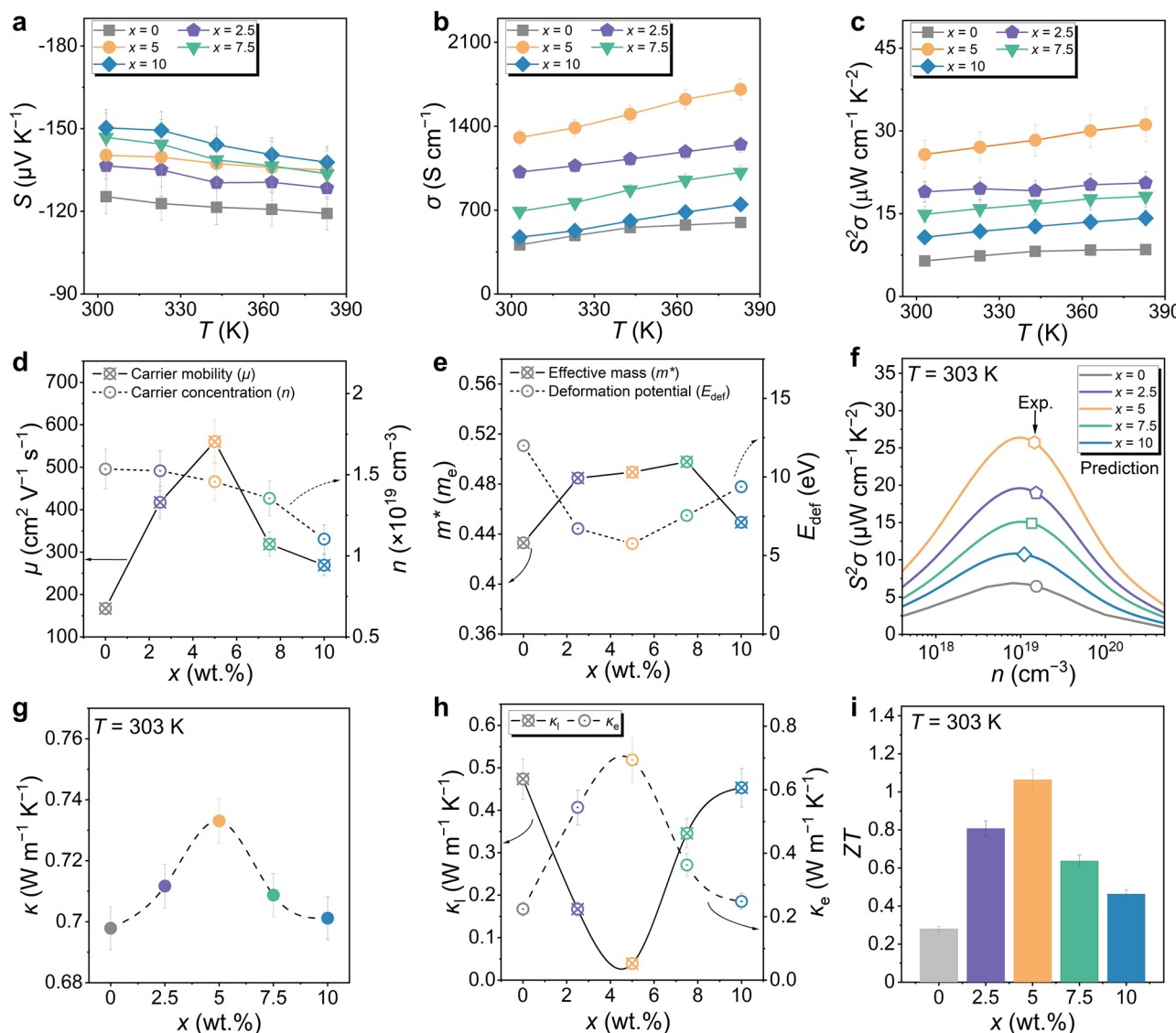

**Fig. 4 | Thermoelectric performance of Ag₂Se films with different Te contents ($x$ = 0, 2.5, 5, 7.5, and 10 wt.%). a** Seebeck coefficient ($S$), **b** electrical conductivity ($\sigma$), and **c** power factor ($S^2\sigma$) as a function of temperature. **d** Measured carrier concentration ($n$) and mobility ($\mu$) at 303 K. **e** Room-temperature effective mass ($m^*$) and deformation potential ($E_{def}$) calculated by the single parabolic band (SPB) model. **f** Comparison of predicted $S^2\sigma$ by the SPB model and measured $S^2\sigma$ as a function of $n$. **g** Room-temperature thermal conductivity ($\kappa$), **h** and corresponding lattice thermal conductivity ($\kappa_l$) and electronic thermal conductivity ($\kappa_e$) as a function of $x$. **i** Comparison of $ZT$ values as a function of $x$ at 303 K.

increases from 0 to 5 wt.%, $\kappa$ rises due to improved densification, as Te fills the pores within the Ag₂Se films. However, when $x$ increases beyond 7.5 wt.%, $\kappa$ declines due to enhanced phonon scattering caused by lattice imperfections and excessive phase boundaries. To distinguish the contributions of $\kappa_e$ and $\kappa_l$, we calculated $\kappa_e$ using the equation $\kappa_e = L\sigma T$ and determined $\kappa_l$ as $\kappa_l = \kappa - \kappa_e$, where the Lorentz number ($L$) was derived from the SPB model (Supplementary Fig. 12). Figure 4h illustrates the variation of $\kappa_e$ and $\kappa_l$ with $x$ at room temperature. From 0 to 5 wt.% Te, $\kappa_e$ increases due to a rise in $\sigma$ associated with enhanced film density, while $\kappa_l$ decreases as stronger phonon scattering occurs from lattice distortions and additional boundaries. Beyond 7.5 wt.% Te, $\kappa_e$ declines as excessive Te phase formation intensifies phonon scattering and reduces $\sigma$. Meanwhile, $\kappa_l$ increases, indicating that the excess Te phase contributes to lattice thermal transport. Figure 4i shows the $ZT$ values as a function of Te content, with a maximum $ZT$ of 1.06 achieved at 303 K for the Ag₂Se film containing 5 wt.% Te. This performance is competitive with previously reported results for screen-printed thermoelectric films.

To assess the flexibility of Ag₂Se films with varying Te content ($x$ = 0, 2.5, 5, 7.5, and 10 wt.%), we conducted a bending test, as shown in Supplementary Fig. 13. Figure 5a shows the normalized resistance change ($\Delta R/R_0$) as a function of bending cycles at a radius ($r$) of 5 mm. The inset depicts the flexible film with $r$ of 5 mm. After 1000 bending cycles, $\Delta R/R_0$ remains below 5% for all samples, demonstrating the excellent flexibility of the as-fabricated Ag₂Se films. To evaluate the practical application potential of our films, we designed an F-TED featuring a triangular structure of p-n junctions. The p-type legs were fabricated using commercial Bi₀.₄Sb₁.₆Te₃ powders through screen printing, the same process used to produce the Ag₂Se films. To enhance the thermoelectric performance of Bi₀.₄Sb₁.₆Te₃, Te was also introduced into the films, with an optimal Te content of 5 wt.%, as determined in our previous study[18]. The thermoelectric properties of Bi₀.₄Sb₁.₆Te₃ with 5 wt.% Te at room temperature are provided in Supplementary Table 3. For the device electrodes, we selected Ag paste and Cu tape based on the band banding concept[17,86]. For the p-type leg (Bi₀.₄Sb₁.₆Te₃), we employed Ag as the contact material. Ag

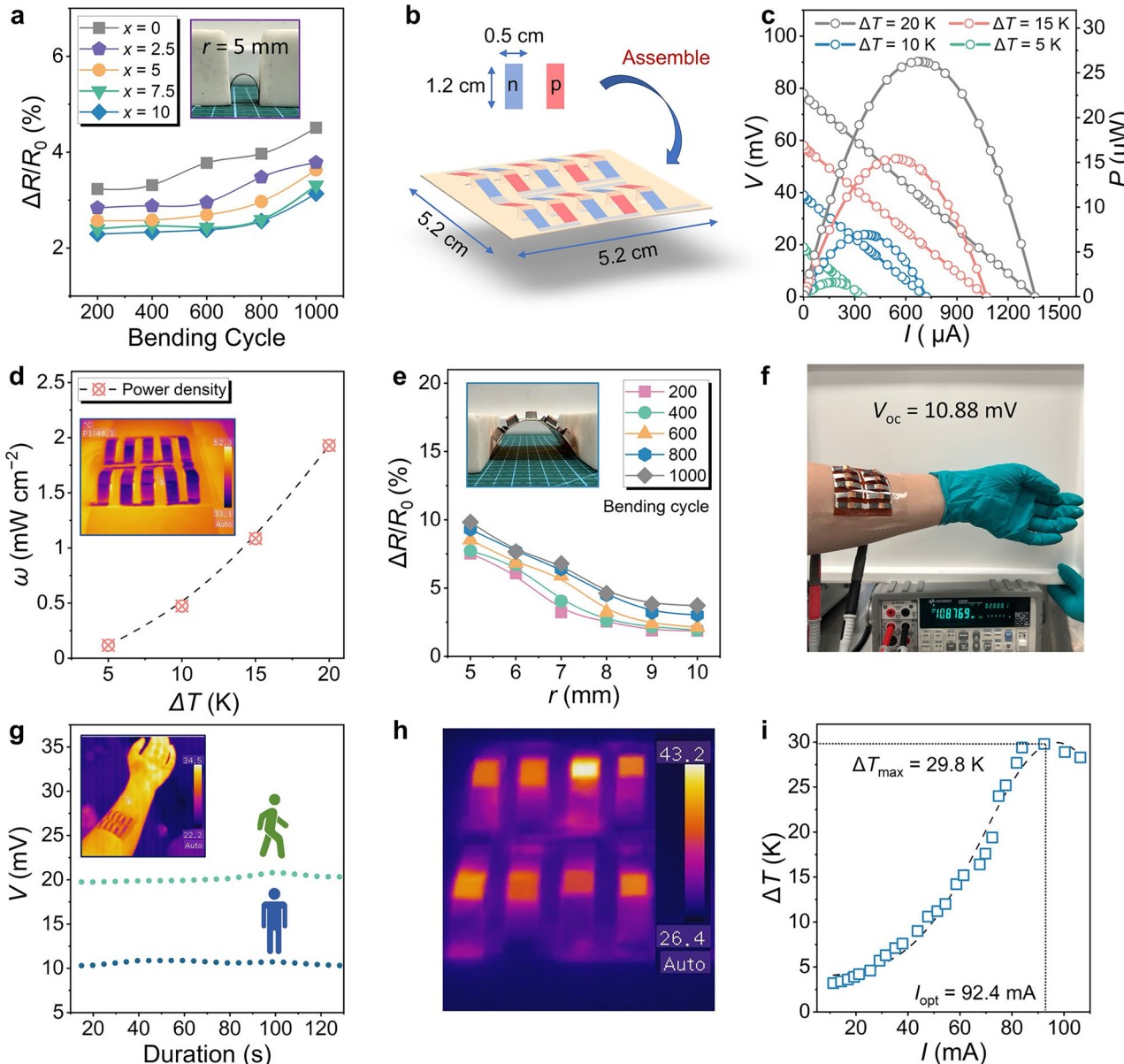

**Fig. 5 | Flexibility of screen-printed Ag₂Se films and performance of as-fabricated device. a** Measured normalized resistance change ($\Delta R/R_0$) of the Ag₂Se thin films with different Te contents ($x = 0, 2.5, 5, 7.5,$ and 10 wt.%) as a function of different bending cycles at 5 mm bending radius ($r$). The inset depicts the flexible film with $r$ of 5 mm. **b** Schematic illustration of the size and structure of as fabricated device. **c** Experimental output voltages ($V$) and output powers ($P$) versus loading current ($I$) with different temperature differences ($\Delta T$s). **d** Determined output power densities ($\omega$) as a function of $\Delta T$. The inset displays the infrared photograph showing the temperature distribution on the as-fabricated device after applying a $\Delta T$. **e** Measured $\Delta R/R_0$ of as-fabricated device versus different $r$ with different bending cycles. The inset depicts the as-fabricated device with a $r$ of 8 mm. **f** Photograph of as-fabricated device worn on a human arm and can provide open-circuit voltage ($V_{oc}$) of 10.88 mV. **g** Experimental $V_{oc}$ of the device worn on a human arm during sitting and walking as a function of time (0 - 130 s). The inset displays an infrared photograph showing the temperature distribution of the device worn on a human arm while seated. **h** The infrared photograph shows the temperature distribution of the device functioning as a cooler after an input current of 69.8 mA is applied. **i** Maximum cooling performance ($\Delta T_{max}$) of the device as a function of input current.

has a work function of 4.3-4.7 eV, which is reasonably close to or higher than the Fermi level of p-type Bi-Sb-Te alloys (typically 4.5-4.7 eV), enabling near-ohmic contact for hole injection[86,87]. For the n-type leg (Ag₂Se), we used Cu, which has a work function of around 4.5 eV[86,88]. This aligns well with the lower Fermi level of n-type Ag₂Se (typically 4.1–4.3 eV), promoting efficient electron injection and minimizing potential Schottky barriers[86]. Additionally, these materials offer low electrical resistance, ease of use, and cost-effectiveness, making them well-suited for flexible thermoelectric applications[17]. Figure 5b illustrates the structure and dimensions of the fabricated device. Its design effectively extends the leg length without increasing vertical space,

enabling a higher $\Delta T$ in compact environments such as electronics. To assess thermal stability, we conducted ANSYS simulations of the temperature distribution (Supplementary Figs. 14 and 15). These results further demonstrate that the device featuring a triangular structure can establish a larger $\Delta T$ compared to the conventional parallel-structured device when heat is applied to the hot side. Figure 5c presents the experimental output voltage ($V$) and power ($P$) as functions of loading current ($I$) at different $\Delta T$s. The maximum open-circuit voltage ($V_{oc}$) and $P$ can reach 77.6 mV and 26.2 μW, respectively, at $\Delta T$ of 20 K. Figure 5d compares the $\omega$ as a function of $\Delta T$. The inset shows an infrared image of the temperature distribution

across the device under an applied $\Delta T$. A competitive $\omega$ of 1.9 mW cm$^{-2}$ is achieved when $\Delta T$ is 20 K and thereby leading to an outstanding $\omega_n$ of 4.8 μW cm$^{-2}$ K$^{-2}$. To evaluate mechanical flexibility, we performed bending tests. Figure 5e shows the measured $\Delta R/R_0$ as a function of $r$ and bending cycles. The inset displays the device under $r$ of 8 mm. Even after 1000 cycles at $r$ of 5 mm, $\Delta R/R_0$ remains below 10%, demonstrating excellent flexibility and stability. The device's potential for wearable electronics is further examined. When worn on a human arm, it generates a $V_{oc}$ of 10.88 mV, as shown in Fig. 5f. Additional dimension details and photographs about the device are shown in Supplementary Figs. 16-17. Additionally, Fig. 5g shows that the device provides a stable $V_{oc}$ when worn on a human arm while walking or sitting, confirming its capability to harness body heat for energy generation. The cooling performance of the device was also evaluated with and without heatsinks, as shown in the Supplementary Figs. 18. Figure 5h presents an infrared image of its temperature distribution when operating as a cooler under an input current of 69.8 mA. A temperature difference of 16.8 K is achieved under this condition. Figure 5i further plots the $\Delta T_{max}$ as a function of input current, showing that $\Delta T_{max}$ reaches 29.8 K at 92.4 mA without external heat sinks. These results confirm that the device is well-suited for various application environments.

In summary, we successfully fabricate high-performance Ag$_2$Se films using screen printing followed by spark plasma sintering. The introduction of Te during fabrication significantly enhanced film density while creating an energy filtering effect that selectively blocked low-energy carriers, leading to increased $S$ and $\sigma$. Additionally, Te diffusion doping occurred during annealing at the Ag$_2$Se-Te boundaries, where Te substitution for Se introduced lattice imperfections, further increasing phonon scattering and reducing $\kappa_l$. As a result, the Ag$_2$Se films with 5 wt.% Te achieved an exceptional $S^2\sigma$ of 25.7 μW cm$^{-1}$ K$^{-2}$ and a competitive $ZT$ of 1.06 at 303 K. The flexibility of the fabricated films was also evaluated. Even after 1000 bending cycles, the $\Delta R/R_0$ remained below 5%, demonstrating excellent mechanical durability. Furthermore, we designed an F-TED incorporating a triangular p-n junction structure, coupling 10 pairs of n-type Ag$_2$Se + 5 wt.% Te films with previously optimized p-type Bi$_{0.4}$Sb$_{1.6}$Te$_3$ + 5 wt.% Te films. This device achieved a $\omega_n$ of 4.8 μW cm$^{-2}$ K$^{-2}$ at a $\Delta T$ of 20 K and provided a $\Delta T_{max}$ of 29.8 K with a current of 92.4 mA. These findings indicate the potential of the fabricated thermoelectric materials and devices for diverse energy-harvesting and cooling applications.

## Methods

### Chemicals
Selenium (Se, 99.99 %, Sigma-Aldrich), silver nitrate (AgNO$_3$, 99 %, Sigma-Aldrich), ethylene glycol (EG, 99 %, Sigma-Aldrich), tellurium dioxide (TeO$_2$, 99.99 %, Sigma-Aldrich) and sodium hydroxide (NaOH, 96 %, Sigma-Aldrich) were used for solvothermal synthesis without further purification. Ethyl cellulose (Sigma-Aldrich), terpineol (Sigma-Aldrich), dibutyl phthalate (Sigma-Aldrich), and dispersant (Disperbyk-110, BYK USA Inc.) were used for fabricating the printing ink. Commercial Bi$_{0.4}$Sb$_{1.6}$Te$_3$ powders (Zhongsheng Heng'an New Material Technology Co., Ltd) were used for fabricating p-type legs.

### Silver selenide and tellurium synthesis
Se and AgNO$_3$ were dissolved in EG (36 ml) with stirring by a magnetic stirrer to form a clear solution. After that, 4 ml NaOH (5 mol L$^{-1}$) was dropped into the solution and formed the precursor solution. Similarly, TeO$_2$ was dissolved in EG, and NaOH was added to the solution. The two prepared solutions were then separately sealed in two 125 ml polytetrafluoroethylene-lined stainless-steel autoclaves. The autoclaves were heated in an oven at 230 °C for 19 h, followed by natural cooling to room temperature. After the synthesis, the two types of products were cooled to room temperature naturally and then

collected by centrifugation and washed with deionized water and ethanol several times. Finally, the washed products were dried in the oven at 60 °C for 24 h.

### The n- and p-type film fabrication and flexibility test
Firstly, the binder solvent was synthesized by mixing ethyl cellulose (10 wt.%), terpineol (80 wt.%), and dibutyl phthalate (10 wt.%). The binder solvent was magnetically stirred for 2 h at 80 °C to produce a clear solution. After that, the solvothermally synthesized powder or commercial p-type powder (80 wt.%) mixed with binder solvent (7.5 wt.%), terpineol (10 wt.%), and dispersant (2.5 wt.%) to prepare printing ink. Next, it was stirred for 2 h to produce a homogeneous viscous solution. The prepared ink was printed onto the flexible polyimide (PI) substrate. The printed films were dried at 200 °C for 0.5 h and annealed in SPS (SPS-211Lx, Fuji Electronic Industrial CO., Ltd., Japan) with 3 MPa at 450 °C for 10 min. The bending test was conducted by simultaneously pushing movable blocks on both sides of the fabricated film, forcing it to bend to a defined radius.

### Flexible device fabrication
The n-type and p-type films are each cut into 10 pairs (Supplementary Fig. 19), with dimensions of 1.2 cm × 0.5 cm, using a laser cutter (TROTEC SPEEDY 300). The resulting pieces are then attached to a PI substrate using silver paste and copper tape. The p-type and n-type elements are positioned at a 30° angle relative to the substrate, while the angle at the p−n junction is 120°.

### Characterization
The morphology and structural characteristics of as-synthesized Ag$_2$Se powder and film were analyzed through SEM (JEOL JSM-7100F, Supplementary Figs. 20, 21) and TEM (HITACHI HF 5000), respectively. The chemical composition and crystal lattice structure of the Ag$_2$Se film were determined by XRD (Bruker-D8) with CuKα radiation over an angular range of 10−80° in 0.02° increments. The chemical valence states were detected using an XPS (Kratos Axis ULTRA, Kratos Analytical Limited, Japan). The element distribution and the chemical composition of the products were determined through EDS mapping and spot analysis (equipped in JEOL JSM-7100F and HITACHI HF 5000).

### Thermoelectric property measurement
The $n$ and $\mu$ values were measured by a Hall system (CH-70, CH-magnetoelectricity Technology Co., Ltd., China) under a magnetic field of up to 500 mT. The ZEM-3 was used to measure the $S$ and $\sigma$ of the Ag$_2$Se film from 303 to 383 K. The $\kappa$ values were calculated by the formula $\kappa = D \times C_p \times \rho$[89], where $D$ is the thermal diffusivity and $C_p$ is the specific heat capacity. $D$ was measured by a laser flash method (LaserPIT, Advance Riko., Inc., Japan) along the in-plane direction. Based on the porosity of the films determined via ImageJ analysis, the $\kappa$ was further corrected using the Maxwell–Eucken model[90]. The $\kappa_e$ was calculated using $\kappa_e = L\sigma T$[89], where $L$ is the Lorenz number. The results were measured more than five times to promise precision.

### Device performance evaluation
A custom-built measurement setup incorporating two multimeters/DC power supply units (KEYSIGHT U3606B and Keithley K2400) was employed to characterize the output voltage and source current of the thermoelectric device. For real-time application measurements, tests were conducted on the wrists of ten volunteers. The $V_{oc}$ was recorded using the KEYSIGHT U3606B multimeter, while the corresponding temperature difference across the device was simultaneously monitored using an infrared thermal camera (see Supplementary Fig. 22).

## Data availability
The data generated in this study is provided in the Source Data file. Source data are provided with this paper.

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

## Acknowledgements

The calculations were conducted in the National Computational Infrastructure, supported by the Australian government, for providing computational resources and services. The authors acknowledge the facilities, and the scientific and technical assistance, of the Australian Microscopy & Microanalysis Research Facility at the Centre for Microscopy and Microanalysis, The University of Queensland. This work was enabled using the Central Analytical Research Facility hosted by the Institute for Future Environments at QUT. Q. Sun acknowledges the Fundamental Research Funds for the Central Universities and Research Funding from West China School/Hospital of Stomatology, Sichuan University, No. QDJF2022-2.

## Author contributions

Z.-G.C. and X.-L.S. supervised the whole project. W. C. prepared materials and measured the thermoelectric properties, conducted scanning electron microscope, designed device structures, fabricated devices, and measured the performance of thermoelectric device. J.Z., M.D., X.-L.S., Q.S., and Z.-G.C. provided data analysis support and financial support. M.L. provided principal calculation support. X.W. provides XRD analysis support. J.O. provided transmission microscopy support. W.C., B.H., T.C., N.L., X.-L.S., W.-D.L., and Z.-G.C. discussed the results. M.Z. and C.Z. provided device assembling and test support. W.C., X.-L.S., and Z.-G.C. analyzed the data and wrote the manuscript. All the authors reviewed and edited the manuscript.

## Competing interests

The authors declare no competing interests.
