## [Transparent Peer Review file · Nature Communications]

Flexible Ag₂Se-based thin-film thermoelectrics for sustainable energy harvesting and cooling

Corresponding Author: Professor Zhi-Gang Chen

Version 0:

Reviewer comments:

Reviewer #1

(Remarks to the Author)

In this present investigation, the authors reported the flexible inorganic thermoelectric materials prepared by screen printing followed by SPS process and obtained a power density of $4.8 \mu\text{W cm}^{-2} \text{K}^{-2}$ at a temperature difference of 19 K and a maximum cooling of 29.8 K with an input current of 92.4 mA. There are some concerns regarding some specific issues.

Comments:

- 1) Since, the Ag₂Se based thermoelectric thin films were studied different synthesis methods such as, hot-rolling, inkjet printing and so on. What are the advantages of screen printing and SPS process?
- 2) How effective the triangular p-n junction from traditional F-TED fabrication methods?
- 3) What is the angle of triangular p-n junction? The detail should be given in experimental section.
- 4) The cooling effect was performed for the F-TED without the use of external heat sink. How the performance will vary with the external heat sink.
- 5) To enhance the discussion on thermal transport properties, the porosity should be measured for all the samples before and after the SPS process.
- 6) The work function detail should be discussed between the p-n legs and the electrodes for the clear understanding. Also, why the authors choose Ag and Cu over other electrode materials?
- 7) How was the bending test performed? The detail should be given in experimental section.
- 8) As an extension of real time applications, the authors are requested to check the thermoelectric device with at least 10 human wrist and compare the temperature difference with performance.

Reviewer #2

(Remarks to the Author)

In this manuscript, Ag₂Se film with Te doped at different wt % has been fabricated and thermoelectric and device output performance has been measured. The film shows a high-power factor of $25.7 \mu\text{W cm}^{-1} \text{K}^{-2}$ and a ZT of 1.06 at 303 K. Additionally, the films show a normalized power density of $4.8 \mu\text{W cm}^{-2} \text{K}^{-2}$ at a temperature difference of 19 K. Te doping in Ag₂Se is not novel; however, the results seem promising. I recommend this manuscript after major revision according to the following points:

1. In Figure S1, only the SEM images of the Te nanorods were provided; however, in the manuscript, Ag₂Se microparticles are also mentioned. The author should provide details of these microparticles' size and the SEM/TEM images to confirm it.
2. There is a typo error in the Result and discussion section: "Figure 1c" should be "Figure 1b".
3. In the manuscript, the Author uses Te nanorods as a composite material with Ag₂Se. Why not Te nanoparticles or Te microparticles? Can they form a better composite with the Ag₂Se material and improve performance?
4. Does the Author conduct relevant repeatability test experiments of solvothermally synthesized Ag₂Se microparticles and Te nanorods?
5. In the manuscript, the Author should mention the thickness and uniformity of the Ag₂Se film fabricated with different Te

content, as Figure S5 does not give a clear hint. Also, to add to this, does the Author measure the transport properties of the fabricated films only at one particular thickness? If yes, the Author should compare the transport properties at different thicknesses for Ag₂Se with (i) 0 wt % Te and (ii) 5 wt % Te.

6. Since the Ag₂Se Films were fabricated on the flexible polyimide (PI) substrate. How does the Author extract the thermal conductivity contribution from the Ag₂Se film part only?

7. Does the Author check the environmental stability of their fabricated film, as silver paint has been used for contacts and Ag paints sometimes crack at some portion after a long time and repeated bending, which can impact the output performance?

8. The Author mentions that Ag₂Se with 5 wt % Te has a normalized power density of 4.8 $\mu\text{Wcm}^{-2}\text{K}^{-2}$ at a temperature difference of 19 K, which is comparable or lower than 5.96 $\mu\text{Wcm}^{-2}\text{K}^{-2}$ for pristine Ag₂Se film at a similar temperature difference and too prepared using the simple vacuum filtration technique (ACS Appl. Mater. Interfaces 2020, 12, 9646–9655). What are the advantages of doping Te to Ag₂Se and preparing the film using the screen-printing techniques if a similar performance can be achieved using a simple vacuum filtration technique? How does the Author justify the performance comparison?

Version 1:

Reviewer comments:

Reviewer #1

(Remarks to the Author)

The authors have made significant efforts to revise the manuscript. However, a few minor comments need to be addressed before the manuscript can be considered for publication.

1. The rationale for choosing tellurium incorporation is not addressed, especially in terms of expected electronic and structural changes

2. The author claimed the porosity percentage is 11% after SPS, which is quite significant; therefore, porosity correction needs to be done to assess the actual thermal conductivity.

Reviewer #2

(Remarks to the Author)

The authors have conducted the relevant experiments and addressed the raised concerns. Thus, I can recommend the paper to be accepted.

Version 2:

Reviewer comments:

Reviewer #1

(Remarks to the Author)

All the raised queries were appropriately addressed with relevant experimental results by the author. I recommend the acceptance of this research article for publication.

Response to reviewers

Reviewer #1: *In this present investigation, the authors reported the flexible inorganic thermoelectric materials prepared by screen printing followed by SPS process and obtained a power density of $4.8 \mu\text{W cm}^{-2} \text{K}^{-2}$ at a temperature difference of 20 K and a maximum cooling of 29.8 K with an input current of 92.4 mA. There are some concerns regarding some specific issues.*

Author reply: We appreciate the reviewer's concerns on this work. We have revised the manuscript to address all the issues you raised.

Comment 1. *Since, the Ag_2Se based thermoelectric thin films were studied different synthesis methods such as, hot-rolling, inkjet printing and so on. What are the advantages of screen printing and SPS process?*

Author reply: Compared to other fabrication methods such as hot-rolling, inkjet printing, and conventional thin-film techniques, screen-printing offers several key advantages for thermoelectric film production—namely cost-effectiveness, scalability, material versatility, and integrated patterning^{1,2}. Screen printing enables film fabrication using relatively low-cost equipment and does not require a vacuum environment, making it a highly accessible and economical technique^{1,3}. Its compatibility with roll-to-roll manufacturing also supports large-scale production, which is essential for practical device integration. In terms of material flexibility, screen-printing inks can be formulated using various thermoelectric materials, such as Bi_2Te_3 -based compounds^{4,5}, Ag_2Se ^{3,6}, and Ag_2Te ⁷, with solvents that are generally compatible across different systems. Furthermore, the technique allows direct patterning of functional device architectures, such as p–n junction arrays, using pre-designed screens^{2,8}. This simplifies the fabrication of film-based thermoelectric devices^{1,8}.

Spark Plasma Sintering (SPS) complements screen printing by addressing some of its limitations, particularly in enhancing the densification of the printed films^{2,4}. SPS can rapidly sinter powder-based films with minimal grain growth, helping to reduce the inherent porosity of screen-printed layers while preserving the nanoscale features and morphologies obtained through methods like solvothermal synthesis^{2,4}. Additionally, the precise temperature control in SPS minimizes undesired reactions or phase transformations, ensuring the retention of the designed composition and microstructure⁹. Compared to conventional thermal annealing, SPS is therefore better suited for sintering screen-printed thermoelectric films derived from nanostructured particles, offering improved microstructural integrity and enhanced functional performance⁴.

Comment 2. *How effective the triangular p-n junction from traditional F-TED fabrication methods?*

Author reply: Most wearable flexible thermoelectric devices (F-TEDs) are designed with a parallel structure, which allows for easy adhesion to human skin. However, this configuration significantly limits the temperature difference between the skin and the surrounding environment, leading to a relatively low open-circuit voltage^{1, 10, 11}.

To address this limitation, enhancing air convection around the thermoelectric (TE) elements is a promising strategy for increasing the temperature difference^{10, 11}. We investigated this approach by comparing a newly designed triangular structure with the conventional parallel F-TED, both under identical dimensions and environmental conditions, as shown in **Fig. R1 (Supplementary Fig. 14** in the Supporting Information). In the traditional parallel structure, the maximum temperature difference achieved was only 0.9 °C. In contrast, the triangular p–n junction design reached a temperature difference of approximately 20 °C. As a result, the triangular structure significantly improves the open-

circuit voltage output compared to the conventional parallel design. We have added a clarification with red in the revised manuscript (Line 22 of Page 16)

Fig. R1. (Supplementary Fig. 14 in the Supporting Information) ANSYS simulations of thermoelectric devices: (a) model of the as-fabricated device with a triangular structure, and (b) its simulated temperature difference. (c) model of a device with a traditional parallel-leg structure, and (d) its corresponding temperature difference. Both device types use thermoelectric legs with identical dimensions. Simulations assume an ambient temperature of 22 °C and a heat convection coefficient of $5 \text{ W m}^{-2} \text{ K}^{-1}$.

Comment 3. *What is the angle of triangular p-n junction? The detail should be given in experimental section.*

Author reply: The p-type and n-type elements are positioned at a 30° angle relative to the substrate, while the angle at the p–n junction is 120° , as illustrated in **Fig. R2** (**Supplementary Fig. 17** in the Supporting Information). The relevant details are provided in the experimental section.

Fig. R2. (**Supplementary Fig. 17** in the Supporting Information) (a) Schematic of the modeled thermoelectric device. (b) Illustration of the angle at the triangular interface of the p–n junction. The model was developed using ANSYS.

Comment 4. *The cooling effect was performed for the F-TED without the use of external heat sink. How the performance will vary with the external heat sink.*

Author reply: The hot side of the fabricated thermoelectric cooler (F-TED) is located at the top p–n junction. To assess the impact of an external heat sink, ten commercial aluminium heat sinks ($7\text{ mm} \times 7\text{ mm} \times 6\text{ mm}$) were attached to the hot side, as shown in **Fig. R3a** (**Supplementary Fig. 18a** in the Supporting Information). **Fig. R3b** (**Supplementary Fig. 18b** in the Supporting Information) presents the maximum temperature difference (ΔT_{max}) of the F-TED with and without heat sinks as a function of input current. When applying a input current of 92.4 mA , the F-TED with heat sinks achieved a ΔT_{max} that was 3.8 K higher than that of the F-TED without heat sinks, due to more efficient heat dissipation from the hot side into the surrounding air^{12,13}. Additionally, the use of heat sinks resulted in a more stable temperature difference during operation, indicating that thermal management on the hot side significantly improves cooling performance^{12,13}. However, the additional volume required for heat sinks may impose limitations on the device's integration in compact systems.

Fig. R3. (Supplementary Fig. 18 in the Supporting Information) (a) Schematic illustration of the dimensions and structure of the commercial heat sinks and the as-fabricated F-TED with the heat sinks attached. (b) Comparison of the maximum temperature difference (ΔT_{\max}) of the device with and without heat sinks as a function of input current.

Comment 5. *To enhance the discussion on thermal transport properties, the porosity should be measured for all the samples before and after the SPS process.*

Author reply: The porosity of the as-fabricated Ag_2Se films containing 0, 2.5, 5, 7.5, and 10 wt.% Te was evaluated using ImageJ software, which analyses SEM images based on area fraction. SEM images and corresponding porosity data for the films prior to spark plasma sintering (SPS) are shown in Fig. R4a-e (Supplementary Fig. 21a-e in the Supporting Information). These results indicate that the initial porosity of the Ag_2Se films is approximately 25%. As the Te content increases, a slight reduction in porosity is observed. This is likely due to the presence of Te nanorods, which tend to fill the voids between Ag_2Se particles. The SEM images and porosity analysis of the Ag_2Se films after SPS, as shown in Fig. R4f-j (Supplementary Fig. 21f-j in the Supporting Information), reveal a

substantial decrease in porosity to approximately 11%. Among the SPS-treated samples, porosity further decreases from 14.87 % to 11.46 % with increasing Te content. These findings demonstrate that both the incorporation of Te nanorods and SPS treatment effectively reduce the porosity of screen-printed Ag_2Se films. This enhanced densification helps explain the transport properties discussed in the article.

Fig. R4. (Supplementary Fig. 21 in the Supporting Information) Scanning electron microscopy (SEM) images and corresponding porosity analysis of Ag₂Se films with 0, 2.5, 5, 7.5, and 10 wt.% Te content **a–e** before SPS and **f–j** after SPS. The porosity analysis is based on ImageJ.

Comment 6. *The work function detail should be discussed between the p-n- legs and the electrodes for the clear understanding. Also, why the authors choose Ag and Cu over other electrode materials?*

Author reply: Thank you for your valuable comment. The work function (ϕ) alignment at the electrode–thermoelectric leg interface is critical for achieving low contact resistance and ensuring efficient charge carrier injection¹⁴. **Fig. R5** presents the mechanism of the energy band after contact of electrode and TE leg. When the ϕ_E is larger than ϕ_S , the energy bands bend upward toward the interface, while the edges bend downward toward the interface when ϕ_E is smaller than ϕ_S ¹⁴. The difference between the ϕ of electrode and TE legs can form barrier at the interface to hinder the carrier transport¹⁴. Therefore, choice of the electrode materials with close ϕ to that of TE legs can reduce the impact of barrier and thereby reducing contact resistance¹⁴. In this study, we choose Bi_{0.4}Sb_{1.6}Te₃ as p-type legs and Ag₂Se as n-type TE legs, hence:

- For the p-type leg (Bi_{0.4}Sb_{1.6}Te₃), we employed Ag as the contact material. Ag has a work function of ~4.3–4.7 eV, which is reasonably close to or higher than the Fermi level of p-type Bi-Sb-Te alloys (typically ~4.5–4.7 eV), enabling near-ohmic contact for hole injection^{14, 15}.
- For the n-type leg (Ag₂Se), we used Cu, which has a work function of ~4.5 eV. This aligns well with the lower Fermi level of n-type Ag₂Se (typically ~4.1–4.3 eV), promoting efficient electron injection and minimizing potential Schottky barriers^{15, 16}.

Furthermore, the choice of Ag and Cu is also based on their excellent electrical conductivity and good chemical compatibility with the respective thermoelectric materials¹⁴. We have added a clarification with red in the revised manuscript (**Line 10 of Page 15**) to explicitly discuss the work function considerations and justify the electrode material selection.

Fig. R5. Energy band diagrams of metal and n-type semiconductor contacts¹⁴. E_{vac} , vacuum energy; E_C , energy of conduction band minimum; E_V , energy of valence band maximum; ϕ_E , electrode work function; ϕ_s , TE legs work function; χ_s , electron affinity of the semiconductor.

Comment 7. How was the bending test performed? The detail should be given in experimental section.

Author reply: The bending test was conducted by simultaneously pushing movable blocks on both sides of the fabricated film, forcing it to bend to a defined radius, as illustrated in **Figs. R6a-b** (**Supplementary Fig. 13a-b** in the Supporting Information). We have added relevant statement in the experimental section.

Fig. R6. (Supplementary Fig. 13 in the Supporting Information) (a) Schematic illustration of the bending test applied to the fabricated film. (b) Photograph of the experimental setup showing the film bent to a radius of 5 mm.

Comment 8. *As an extension of real time applications, the authors are requested to check the thermoelectric device with at least 10 human wrist and compare the temperature difference with performance.*

Author reply: Thank you very much for your constructive suggestion. Following your advice, we conducted additional tests on 10 volunteers to evaluate the open-circuit voltage (V_{oc}) output of the thermoelectric device when worn on the human wrist during both walking and sitting conditions. As shown in **Fig. R7 (Supplementary Fig. 22 in the Supporting Information)**, the device exhibited stable V_{oc} output across different individuals. Infrared thermal imaging was used to characterize the temperature distribution between the skin and ambient environment. The skin temperature of the participants ranged from 30.4 °C to 33.8 °C. For the device worn on the human wrist, the hot-side temperature was approximately 28.3 °C, while the cold-side temperature was around 25.5 °C. The ambient temperature during the measurements was approximately 22.4 °C. As a result, the effective temperature difference across the device was similar among individuals. Under sitting conditions, the device generated a V_{oc} ranging from 8.49 mV to 11.48 mV, while under walking conditions, the V_{oc} increased to 17.46 mV to 20.99 mV due to enhanced heat dissipation. These results demonstrate that the device can consistently produce thermoelectric output across different users and under varying activity states, confirming its potential for real-world wearable applications.

Fig. R7. (Supplementary Fig. 22 in the Supporting Information) Experimental V_{oc} of the device worn on 10 different human arms during sitting and walking as a function of time (0~130 s). The inset displays an infrared photograph showing the temperature distribution of the device worn on the human arms while seated.

Reviewer #2: *In this manuscript, Ag₂Se film with Te doped at different wt % has been fabricated and thermoelectric and device output performance has been measured. The film shows a high-power factor of 25.7 $\mu\text{W cm}^{-1}\text{K}^{-2}$ and a ZT of 1.06 at 303 K. Additionally, the films show a normalized power density of 4.8 $\mu\text{W cm}^{-2}\text{K}^{-2}$ at a temperature difference of 20 K. Te doping in Ag₂Se is not novel; however, the results seem promising. I recommend this manuscript after major revision according to the following points:*

Author reply: We appreciate the reviewer's positive feedback on this work. We have revised the manuscript to address all the issues you raised.

Comment 1. *In Figure S1, only the SEM images of the Te nanorods were provided; however, in the manuscript, Ag₂Se microparticles are also mentioned. The author should provide details of these microparticles' size and the SEM/TEM images to confirm it.*

Author reply: The morphology and size of the synthesized Ag₂Se microparticles are shown in **Fig. R8 (Supplementary Fig. 1** in the Supporting Information). The particles exhibit irregular shapes, with an average diameter of approximately 10–20 μm . We have added the relevant SEM images on **page 6** of revised Supporting Information.

Fig. R8. (Supplementary Fig. 1 in the Supporting Information) Scanning electron microscopy (SEM) images of Ag₂Se microparticles with (a) low- and (b) high-magnification.

Comment 2. *There is a typo error in the Result and discussion section: "Figure 1c" should be "Figure 1b".*

Author reply: We have corrected the typo in the result and discussion section as: “This diffusion doping increases the Ag₂Se bandgap from 0.07 to 0.127 eV, further improving n and S . As a result, the Ag₂Se/Te film achieves an exceptional $S^2\sigma$ of 25.7 $\mu\text{W cm}^{-1} \text{K}^{-2}$ at 303 K, the highest reported for screen-printed films (**Fig. 1b**)^{4, 6, 7, 17-31}.” For your convenience, we have marked the corrected typo as red on **Line 16** of **Page 5** in the revised manuscript.

Comment 3. *In the manuscript, the Author uses Te nanorods as a composite material with Ag₂Se. Why not Te nanoparticles or Te microparticles? Can they form a better composite with the Ag₂Se material and improve performance?*

Author reply: Te nanorods can be synthesized without the use of surfactants, which are typically employed in solvothermal methods to control crystal growth and morphology^{32, 33}. While surfactants such as polyvinylpyrrolidone (PVP) are effective in tailoring the size and shape of nanocrystals, they are often difficult to fully remove^{32, 33}. Residual surfactants can remain on the particle surfaces, leading to unwanted impurities that hinder thermoelectric performance, particularly by reducing electrical conductivity (σ)^{32, 33}. In this study, Te was used as a filler material to bridge the gaps between Ag₂Se microparticles. The presence of surfactant residues on Te nanoparticles would compromise both the interfacial binding and the electrical conductivity of the composite. To avoid these issues, Te nanorods synthesized without surfactants were selected as the filler, offering a cleaner interface and improved performance.

Comment 4. *Does the Author conduct relevant repeatability test experiments of solvothermally synthesized Ag_2Se microparticles and Te nanorods?*

Author reply: To evaluate the reproducibility of our solvothermal synthesis method, we repeated the preparation of Ag_2Se microparticles and Te nanorods three times using identical reactants and reaction conditions. The morphology and size of the Ag_2Se microparticles from the repeated experiments are shown in **Figs. R9a-c (Supplementary Figs. 20a-c of the Supporting Information)**. The results consistently reveal irregularly shaped microparticles with an average diameter of 10–20 μm . Similarly, the morphology and dimensions of the Te nanorods, presented in **Figs. R9d-e (Supplementary Fig. 20d-e)**, remained consistent across all trials. These findings demonstrate the high reproducibility of the solvothermal synthesis process for both Ag_2Se microparticles and Te nanorods.

Fig. R9. (Supplementary Fig. 20 in the Supporting Information) Scanning electron microscopy (SEM) images of (a)-(c) Ag_2Se microparticles and (d)-(e) Te nanorods synthesized in three independent trials, demonstrating consistent morphology and size across repetitions.

Comment 5. *In the manuscript, the Author should mention the thickness and uniformity of the Ag₂Se film fabricated with different Te content, as Figure S5 does not give a clear hint. Also, to add to this, does the Author measure the transport properties of the fabricated films only at one particular thickness? If yes, the Author should compare the transport properties at different thicknesses for Ag₂Se with (i) 0 wt % Te and (ii) 5 wt % Te.*

Author reply: Thank you very much for your insightful and constructive comment. In the revised manuscript, we have clarified the thickness and uniformity of Ag₂Se films with different Te content with red as “The film thickness and uniformity show only slight variation (12–16 μm) with different Te content since all films were fabricated using 160 mesh screens. This consistency is attributed to two factors: Firstly, the Te content in the precursor is relatively low compared to Ag₂Se. Secondly, all films were annealed under the same spark plasma sintering (SPS) conditions, which involved high temperature and pressure, ensuring uniform densification and minimal influence of additional Te on film morphology.” On **Line 7** of **Page 7** in the revised manuscript.

Furthermore, to investigate the effect of thickness on the transport properties, we prepared additional Ag₂Se films with 0 wt% and 5 wt% Te using 110 mesh screens, which produced thicker films (~30 μm), as shown in **Fig. R10a**, confirming the uniformity of these thicker films. The corresponding transport properties: Seebeck coefficient (S), electrical conductivity (σ), and power factor ($S^2\sigma$) are compared in **Figs. R10b-d** between the 15 μm and 30 μm thick films. The results show that S remains nearly unchanged with film thickness but σ increases slightly in thicker films, and thereby $S^2\sigma$ shows a modest increase with thickness. These findings support the robustness of our conclusions regarding Te doping effects, independent of minor variations in film thickness. They also highlight a key advantage of the screen-printing technique, which is its excellent control over film

thickness through the selection of screen mesh size. We sincerely appreciate this comment, which has helped us strengthen the manuscript and provide a more comprehensive view of the material properties.

Fig. R10. (a) Scanning electron microscopy (SEM) images of Ag_2Se films ($30 \mu\text{m}$) with $x = 0$ and $x = 5 \text{ wt.}\% \text{ Te}$ from cross-sectional views. Comparison of (b) Seebeck coefficient (S), (c) electrical conductivity (σ), and (d) power factor ($S^2\sigma$) of Ag_2Se films with thickness of $15 \mu\text{m}$ and $30 \mu\text{m}$ as a function of temperature.

Comment 6. *Since the Ag_2Se Films were fabricated on the flexible polyimide (PI) substrate. How does the Author extract the thermal conductivity contribution from the Ag_2Se film part only?*

Author reply: We utilized LaserPIT to measure the in-plane thermal diffusivity (D) of both the Ag₂Se films on PI substrates and the pure PI substrate. The corresponding thermal conductivities were then calculated using the relation $\kappa = D \times C_p \times \rho$ for each sample³⁴. To estimate the intrinsic in-plane thermal conductivity (κ_{film}) of the Ag₂Se film itself, we employed the rule of mixtures, also known as the Voigt model, which is commonly used in materials science to describe composite behavior under parallel heat flow conditions. The Voigt model for the in-plane configuration is given as³⁵⁻³⁷:

$$D_{eff} = \sum_i v_i \cdot D_i \quad (1)$$

$$D_{total} = \frac{t_{film}}{t_{total}} \cdot D_{film} + \frac{t_{sub}}{t_{total}} \cdot D_{sub} \quad (2)$$

Solving for κ_{film} :

$$D_{film} = \frac{t_{total}}{t_{film}} \cdot D_{total} + \frac{t_{sub}}{t_{film}} \cdot D_{sub} \quad (3)$$

$$\kappa_{film} = D_{film} \cdot C_p \cdot \rho_{film} \quad (4)$$

D_{eff} is measured thermal diffusivity, v_i is the volume fraction for each layer, and D_i is the thermal diffusivity of each layer. D_{total} is measured in-plane thermal diffusivity of Ag₂Se film with PI substrate, D_{film} is in-plane thermal diffusivity of Ag₂Se film, D_{sub} is in-plane thermal diffusivity of PI substrate. t_{total} is thickness of Ag₂Se film with PI substrate, t_{film} thickness of Ag₂Se film, t_{sub} is thickness of PI substrate.

Comment 7. *Does the Author check the environmental stability of their fabricated film, as silver paint has been used for contacts and Ag paints sometimes crack at some portion after a long time and repeated bending, which can impact the output performance?*

Author reply: For environmental stability, we conducted repeated bending tests with varying bending radii, as shown in **Fig. R11** (corresponding to **Fig. 5e** in the manuscript). The results demonstrate that

even after 1,000 cycles at a bending radius of 5 mm, the relative resistance change ($\Delta R/R_0$) remains below 10%, indicating good mechanical flexibility and stability. However, the interface between the Ag paste and the thermoelectric (TE) legs may be prone to cracking under repeated thermal cycling at elevated temperatures or in humid environments due to moisture absorption. To enhance long-term stability, encapsulating materials such as polydimethylsiloxane (PDMS), known for its flexibility and protective properties, can be employed to shield the Ag paste from mechanical and environmental stress. Nevertheless, selecting an appropriate encapsulant requires extensive experimentation to ensure proper adhesion to the Ag paste after post-processing and curing. In this study, our primary objective was to demonstrate a high-performance flexible thermoelectric device (F-TED) with triangular p-n junctions. Despite the absence of encapsulation, the fabricated device maintained good stability after 1,000 bending cycles at a 5 mm radius. While encapsulation strategies are beyond the scope of this work, they will be explored in future studies.

Fig. R11. (Fig. 5e in the manuscript) Measured $\Delta R/R_0$ of as-fabricated device versus different r with different bending cycles. The inset depicts the as-fabricated device with a r of 8 mm.

Comment 8. *The Author mentions that Ag₂Se with 5 wt % Te has a normalized power density of 4.8 $\mu\text{Wcm}^{-2}\text{K}^{-2}$ at a temperature difference of 20 K, which is comparable or lower than 5.96 $\mu\text{Wcm}^{-2}\text{K}^{-2}$ for pristine Ag₂Se film at a similar temperature difference and too prepared using the simple vacuum filtration technique (ACS Appl. Mater. Interfaces 2020, 12, 9646–9655). What are the advantages of doping Te to Ag₂Se and preparing the film using the screen-printing techniques if a similar performance can be achieved using a simple vacuum filtration technique? How does the Author justify the performance comparison?*

Author reply: We sincerely thank the reviewer for the insightful comment. After a thorough review of the work reported in ACS Appl. Mater. Interfaces 2020, 12, 9646–9655, we acknowledge that the pristine Ag₂Se films fabricated by vacuum filtration exhibit a commendable normalized power density of 5.96 $\mu\text{W cm}^{-2} \text{K}^{-2}$. However, a closer performance comparison reveals that our screen-printed Te-doped Ag₂Se films demonstrate higher electrical conductivity, which plays a crucial role in overall power output.

This improvement can be primarily attributed to two factors: (i) Te incorporation and (ii) spark plasma sintering (SPS) post-treatment. The addition of Te, a metallic element, enhances carrier transport by bridging inter-particle interfaces, thus facilitating improved electrical conduction. Moreover, the SPS process enables rapid densification at relatively low temperatures while preserving the microstructure, effectively avoiding the grain coarsening often seen in traditional hot pressing (as used in the referenced study)^{5,9}. This helps maintain the nanoscale features crucial to thermoelectric performance^{4,5}. In addition to material enhancements, our screen-printing method offers practical advantages over vacuum filtration for scalable and flexible device fabrication. Screen printing allows direct deposition on various substrates, precise control over film thickness, and patterning of defined geometries, making it highly suitable for manufacturing large-area thermoelectric modules with distinct n-type and p-type legs^{1,2}. In contrast, vacuum filtration is inherently limited in film uniformity control, substrate compatibility, and industrial scalability¹⁴.

In summary, although both methods yield comparable normalized power density values, our approach integrates material optimization (Te doping + SPS) with a scalable, industry-relevant fabrication technique (screen printing). This combination not only enhances electrical conductivity but also enables better control over film architecture and device integration, highlighting the practical advantages of our method in real-world thermoelectric applications. The comparison details can be found in **Table R1**:

Table R1. Summary of advantages in this work compared with Ref. *ACS Appl. Mater. Interfaces* 2020, 12, 9646–9655.

	This work (Screen Printing)	Ref. (Vacuum Filtration)
Post-treatment Method	Spark Plasma Sintering (SPS)- 5min	Conventional Hot Pressing- 30min
Grain Morphology	Fine grains preserved via rapid sintering	Risk of grain coarsening due to prolonged heating
Power Factor	25.7 $\mu\text{W cm}^{-1} \text{K}^{-2}$	18.82 $\mu\text{W cm}^{-1} \text{K}^{-2}$
Patterning Capability	Yes: Direct printing of n/p legs	No: Not suitable for selective patterning
Scalability & Throughput	High: can be used for roll-to- roll printing	Low: Lab-scale only

Reference

1. Zhang X, *et al.* Stamp-Like Energy Harvester and Programmable Information Encrypted Display Based on Fully Printable Thermoelectric Devices. *Adv Mater* **35**, 2207723 (2023).
2. Chen W, *et al.* Nanobinders Advance Screen-Printed Flexible Thermoelectrics. *Science* **386**, 1265-1271 (2024).
3. Zhang M, *et al.* Screen printing Ag₂Se/carbon nanocomposite films for flexible thermoelectric applications. *Carbon* **229**, 119480 (2024).
4. Shi J, *et al.* Anisotropy engineering in solution-derived nanostructured Bi₂Te₃ thin films for high-performance flexible thermoelectric devices. *Chem Eng J* **458**, 141450 (2023).
5. Varghese T, *et al.* Flexible Thermoelectric Devices of Ultrahigh Power Factor by Scalable Printing and Interface Engineering. *Adv Funct Mater* **30**, 1905796 (2020).
6. Zhang M, *et al.* Scalable printing high-performance and self-healable Ag₂Se/terpineol nanocomposite film for flexible thermoelectric device. *Energy* **296**, 131232 (2024).
7. Du J, *et al.* Inkjet Printing Flexible Thermoelectric Devices Using Metal Chalcogenide Nanowires. *Adv Funct Mater* **33**, 2213564 (2023).
8. Wang H, *et al.* Flexible Porous Ag₂Se Films: From Freestanding Inorganic Films to Inorganic-Network/Organic-Skeleton Thermoelectric Generators. *Adv Funct Mater* **35**, 2413605 (2025).
9. Shi X-L, Zou J, Chen Z-G. Advanced Thermoelectric Design: From Materials and Structures to Devices. *Chem Rev* **120**, 7399-7515 (2020).
10. Cao T, *et al.* Advances in bismuth-telluride-based thermoelectric devices: progress and challenges. *eScience* **3**, 100122 (2023).

11. Zheng Z-H, *et al.* Harvesting waste heat with flexible Bi₂Te₃ thermoelectric thin film. *Nat Sustain* **6**, 180-191 (2023).
12. Mao J, Chen G, Ren Z. Thermoelectric cooling materials. *Nat Mater* **20**, 454-461 (2021).
13. Zhou L, Meng F, Sun Y. Numerical study on infrared detectors cooling by multi-stage thermoelectric cooler combined with microchannel heat sink. *Appl Therm Eng* **236**, 121788 (2024).
14. Zhang Z, Yates JT, Jr. Band Bending in Semiconductors: Chemical and Physical Consequences at Surfaces and Interfaces. *Chem Rev* **112**, 5520-5551 (2012).
15. Kumar B, Kaushik BK, Negi YS. Perspectives and challenges for organic thin film transistors: materials, devices, processes and applications. *J Mater Sci-Mater El* **25**, 1-30 (2014).
16. Brodie I, Chou SH, Yuan H. A general phenomenological model for work function. *Surf Sci* **625**, 112-118 (2014).
17. Xie J, *et al.* Flexible pCu₂Se-nAg₂Se thermoelectric devices *via in situ* conversion from printed Cu patterns. *Chem Eng J* **435**, 135172 (2022).
18. Mallick MM, *et al.* High Figure-of-Merit Telluride-Based Flexible Thermoelectric Films through Interfacial Modification *via* Millisecond Photonic-Curing for Fully Printed Thermoelectric Generators. *Adv Sci* **9**, 2202411 (2022).
19. Chen B, *et al.* Flexible thermoelectric generators with inkjet-printed bismuth telluride nanowires and liquid metal contacts. *Nanoscale* **11**, 5222-5230 (2019).

20. Mallick MM, *et al.* New frontier in printed thermoelectrics: formation of β -Ag₂Se through thermally stimulated dissociative adsorption leads to high ZT. *J Mater Chem A* **8**, 16366-16375 (2020).
21. Yuan Z, *et al.* High-Performance Micro-Radioisotope Thermoelectric Generator with Large-Scale Integration of Multilayer Annular Arrays through Screen Printing and Stacking Coupling. *Energy Technol* **9**, 2001047 (2021).
22. Feng J, Zhu W, Deng Y, Song Q, Zhang Q. Enhanced Antioxidation and Thermoelectric Properties of the Flexible Screen-Printed Bi₂Te₃ Films through Interface Modification. *ACS Appl Energy Mater* **2**, 2828-2836 (2019).
23. Feng J, Zhu W, Zhang Z, Cao L, Yu Y, Deng Y. Enhanced Electrical Transport Properties via Defect Control for Screen-Printed Bi₂Te₃ Films over a Wide Temperature Range. *ACS Appl Mater Interfaces* **12**, 16630-16638 (2020).
24. Mallick MM, *et al.* High-Performance Ag - Se-Based n-Type Printed Thermoelectric Materials for High Power Density Folded Generators. *ACS Appl Mater Interfaces* **12**, 19655-19663 (2020).
25. Liu Y, *et al.* Fully inkjet-printed Ag₂Se flexible thermoelectric devices for sustainable power generation. *Nat Commun* **15**, 2141 (2024).
26. Zhu W, Deng Y, Gao M, Wang Y. Hierarchical Bi - Te based flexible thin-film solar thermoelectric generator with light sensing feature. *Energ Convers Manage* **106**, 1192-1200 (2015).

27. Cao Z, Koukharenko E, Tudor MJ, Torah RN, Beeby SP. Flexible screen printed thermoelectric generator with enhanced processes and materials. *Sensor Actuat A-phys* **238**, 196-206 (2016).
28. Chen B, *et al.* Inkjet Printing of Single-Crystalline Bi₂Te₃ Thermoelectric Nanowire Networks. *Adv Electron Mater* **3**, 1600524 (2017).
29. Gao J, *et al.* A Novel Glass-Fiber-Aided Cold-Press Method for Fabrication of *n*-Type Ag₂Te Nanowires Thermoelectric Film on Flexible Copy-Paper Substrate. *J Mater Chem A* **5**, 24740-24748 (2017).
30. Varghese T, *et al.* High-Performance and Flexible Thermoelectric Films by Screen Printing Solution-Processed Nanoplate Crystals. *Sci Rep* **6**, 33135 (2016).
31. Chang P-S, Liao C-N. Screen-printed flexible thermoelectric generator with directional heat collection design. *J Alloys Compd* **836**, 155471 (2020).
32. Chen Y, *et al.* Facile surfactant-free microwave-assisted solvothermal synthesis of Cu₂Te_{1-x}S_x with enhanced thermoelectric performance. *J Eur Ceram Soc* **44**, 6374-6383 (2024).
33. Ibrahim EMM, *et al.* Effect of surfactant concentration on the morphology and thermoelectric power factor of PbTe nanostructures prepared by a hydrothermal route. *Physica E* **125**, 114396 (2021).
34. Wang Y, *et al.* Enhanced Thermoelectric Properties of Nanostructured *n*-Type Bi₂Te₃ by Suppressing Te Vacancy through Non-Equilibrium Fast Reaction. *Chem Eng J* **391**, 123513 (2019).

35. Tian W, Fu MW, Qi L, Ruan H. Micro-mechanical model for the effective thermal conductivity of the multi-oriented inclusions reinforced composites with imperfect interfaces. *Int J Heat Mass Tran* **148**, 119167 (2020).
36. Zhao L-D, Dravid VP, Kanatzidis MG. The Panoscopic Approach to High Performance Thermoelectrics. *Energy Environ Sci* **7**, 251-268 (2014).
37. Wang J, Carson JK, North MF, Cleland DJ. A new approach to modelling the effective thermal conductivity of heterogeneous materials. *Int J Heat Mass Tran* **49**, 3075-3083 (2006).

Response to reviewers

Reviewer #1: *The authors have made significant efforts to revise the manuscript. However, a few minor comments need to be addressed before the manuscript can be considered for publication.*

Author reply: We appreciate the reviewer's concerns on this work. We have revised the manuscript to address all the issues you raised.

Comment 1. *The rationale for choosing tellurium incorporation is not addressed, especially in terms of expected electronic and structural changes?*

Author reply: We agree that the rationale for tellurium incorporation should be more clearly elaborated. In our study, tellurium was introduced into the Ag₂Se films for several reasons related to both structural and electronic enhancements: First, tellurium acts as a nanobinder during the film formation process, effectively connecting neighboring Ag₂Se particles. This contributes to improved film densification, which in turn enhances electrical conductivity (σ). The phenomenon of enhancing the film densification has been reported in the previous work¹⁻³. Second, Te has a larger atomic radius and lower electronegativity compared to Se or S⁴. When Te is substituted for Se, it modifies the local electronic environment and affects the band structure⁴. As a result, the bandgap of Te-doped Ag₂Se increases to 0.127 eV compared to 0.07 eV in pristine Ag₂Se, as shown in **Fig. R1a-b** (corresponding to **Fig. 1a** in the manuscript). The results are based on first-principles DFT calculations. This wider bandgap facilitates improved Seebeck coefficient (S) due to suppressed bipolar conduction and better energy filtering of carriers⁴. Third, the incorporation of Te into the Se sublattice leads to lattice expansion due to larger ionic radius of Te. This structural change is confirmed by the Rietveld refinement of the XRD data, as shown in **Fig. R1c** (corresponding to **Fig. 2c** in the manuscript). The

expanded lattice introduces additional phonon scattering centers, thereby reducing the lattice thermal conductivity (κ_l), which is beneficial for enhancing the overall thermoelectric figure of merit (ZT). In summary, Te incorporation plays a synergistic role in optimizing the microstructure, electronic band structure, and phonon transport of Ag_2Se film, which collectively contribute to the improved thermoelectric performance observed in our work.

Fig. R1. (Fig. 1a and Fig. 2c in the manuscript) **Schematic illustration of the mechanism and structure of screen-printed $\text{Ag}_2\text{Se}/\text{Te}$ films.** The calculated bandgap of (a) pristine Ag_2Se and (b) Te-doped Ag_2Se . The results are based on the first-principles DFT calculations.

Comment 2. *The author claimed the porosity percentage is 11% after SPS, which is quite significant; therefore, porosity correction needs to be done to assess the actual thermal conductivity?*

Author reply: We would like to clarify that the thermal conductivity values reported in the manuscript have already been corrected for porosity, as shown in Fig. R2 (corresponding to Fig. 4g in the manuscript). However, we acknowledge that the methodology and parameters used for this correction were not explicitly stated in the original version, which may have led to confusion. In this work, we

employed the Maxwell–Eucken model to estimate the intrinsic thermal conductivity of the porous Ag₂Se films. This model is commonly used to evaluate the effective thermal conductivity of heterogeneous systems, particularly when non-conductive pores are dispersed within a continuous matrix⁵. The corresponding equation is as follows⁵:

$$\kappa_{eff} = \frac{\kappa_m v_m}{v_m + \frac{3}{2} v_p} \quad (1)$$

where κ_m is the thermal conductivity of the dense Ag₂Se matrix; κ_p is the thermal conductivity of the pore phase (assumed to be zero); v_m and v_p represent the volume fractions of the matrix and pores, respectively. In our case, a non-conductive pore model was adopted given that the pores are assumed to be air-filled. The porosity of the films with different Te content were estimated from SEM images using ImageJ analysis. Details of the correction method and the relevant parameters have now been added to the experimental section of the Supplementary Information for clarity and reproducibility. We appreciate the reviewer’s suggestion, which helped us improve the transparency and scientific rigor of our thermal conductivity analysis.

Fig. R2. (Fig. 4g in the manuscript) Room-temperature thermal conductivity (κ) as a function of x .

Reviewer #2: *The authors have conducted the relevant experiments and addressed the raised concerns.*

Thus, I can recommend the paper to be accepted.

Author reply: We appreciate the reviewer's positive response on this work.

References

1. Chen W, *et al.* Nanobinders Advance Screen-Printed Flexible Thermoelectrics. *Science* **386**, 1265-1271 (2024).
2. Shi J, *et al.* Anisotropy engineering in solution-derived nanostructured Bi₂Te₃ thin films for high-performance flexible thermoelectric devices. *Chem Eng J* **458**, 141450 (2023).
3. Varghese T, *et al.* Flexible Thermoelectric Devices of Ultrahigh Power Factor by Scalable Printing and Interface Engineering. *Adv Funct Mater* **30**, 1905796 (2020).
4. Yang D, *et al.* Flexible power generators by Ag₂Se thin films with record-high thermoelectric performance. *Nat Commun* **15**, 923 (2024).
5. Wang J, Carson JK, North MF, Cleland DJ. A new approach to modelling the effective thermal conductivity of heterogeneous materials. *Int J Heat Mass Tran* **49**, 3075-3083 (2006).